

**Estimating distributed soil texture using time series of thermal**
**remote sensing – A case study in central Europe**
B. Müller[1,2], M. Bernhardt[1], C. Jackisch[3], K. Schulz[1]
[1]Institute of Water Management, Hydrology and Hydraulic Engineering, University of Natural Resources and Life
Sciences, Vienna, Austria
[2]Department of Geography, Ludwig-Maximilians-Universität, Munich, Germany
[3]Institute of Water and River Basin Management, Karlsruhe Institute of Technology, Karlsruhe, Germany
*Correspondence to:* Matthias Bernhardt (matthias.bernhardt@boku.ac.at)
**Abstract**. For understanding water and solute transport processes knowledge about the respective hydraulic
properties is necessary. Commonly, hydraulic parameters are estimated via pedo-transfer functions using soil
texture data to avoid cost intensive measurements of hydraulic parameters in the laboratory. Therefore, current
soil texture information is only available at coarse spatial resolution of 250 m to1000 m. Here, a method is
presented to derive high-resolution (15 m) topsoil texture patterns for the meso-scale Attert catchment
(Luxembourg, 288 km²) from 28 images of ASTER thermal remote sensing. A principle component analysis of
the images reveals the most dominant thermal patterns (principle components, PCs) that are related to 212
fractional soil texture samples. Within a multiple linear regression framework, distributed soil texture information
is estimated and related uncertainties are assessed. An overall root mean squared error of 12.7 percentage points
(pp) lies well within and even below the range of recent studies on soil texture estimation, while requiring sparser
sample setups and a less diverse set of basic spatial input.
**1    Introduction**
The prediction of (sub-)surface water and solute transport processes, from the plot to the basin scale, heavily rely
on spatial information of soil hydraulic properties (SHPs). However, the measurement of SHPs in the field or in
the lab is very time consuming and expensive with regard to equipment and labor costs (Durner and Lipsius, 2005).
To reduce experimental efforts and cost, SHPs are often estimated via so-called pedotransfer-functions from other
soil properties that are easier and cheaper available; examples of relevant properties are soil texture, bulk density
or organic carbon content (Pachepsky and Rawls, 2004).
Standard soil texture information, from country to global levels, is available from a variety of sources. They vary
in resolution, method of production and quality. Exemplary soil texture sources on country to global level are JRC
(2015), FAO (2015), ISRIC (2015), USDA (2015), or AAFC (2015). The spatial resolutions of these products
differ between 250 m and 1000 m. The product quality is defined by usually sparse sample data and additional
spatial information ranging from topography, landform observations, remote sensing products, or expert
knowledge. Also applied interpolation techniques and landscape evolution models, as well as pattern estimation
methods control the quality of derived spatial soil texture products. In general, texture information have a coarse
spatial resolution, and partly consist only of qualitative information, thus are characterized by large uncertainties
which are hard to quantify.





However, many current applications, e.g. land surface models, heavily rely on high-resolution products in the
range of 100 m and less (precision farming: 10-50 m, e.g. Selige et al., 2006, or Sadler et al., 1998; flood
forecasting: 50-100 m, e.g. de Roo et al., 2003, or Reed et al., 2007; etc.). Any quantification of uncertainties with
regard to map information is missing most of the time, but would be essential in order to evaluate the quality and
reliability of model predictions.
This study uses thermal remote sensing (RS) data in combination with plot measurements to generate spatially
distributed soil texture maps. Physical relations between surface temperature, thermal radiation, soil water content,
and soil texture have been widely demonstrated in studies to determine soil texture characterizations (diurnal
temperature range: Wang et al., 2015; multispectral data: Ahmed and Iqbal, 2014; partial regressions with thermal
spectra: Dhawale, 2015). If time series of thermal RS data are available, the concept of thermal inertia is applicable
to gain information on soil texture. Thermal inertia is the spatially varying tendency of the land surface to resist
changes in temperature forced by energy input. Responsible for these spatial differences in inertia are patterns of
thermal conductivity, density, and specific heat capacity of the land surface material (Rees and Rees, 2013;
Minacapilli et al., 2012). However, thermal observations of land surface are non-linear integrals over all three
dimensions in space of the occurring materials (Hall et al., 1995; Betts et al., 1996). These integrals consider spatial
averaging, as well as thermal emission and propagation from sub surface thermal sources up to vegetation.
Parameters that influence the surface temperature are incoming radiation, land use, albedo, and available water
content. Especially the latter is strongly controlled by soil texture, which subsequently should influence the thermal
inertia signature as given by the temporal patterns of surface temperature.
In a previous investigation, Müller et al. (2014) have utilized principal component analysis (PCA) for a statistical
extraction of dominant patterns within an ASTER thermal infrared (TIR) time series. The produced principal
components (PCs) are independent and will be used here to derive stable patterns that are related to soil texture
classes. A multi-linear regression estimator (MLRE) is used in this context. The MLRE is able to establish and
estimate a functional relationship between the PCs and the fractional texture information from multiple soil texture
samples within a catchment. The resulting spatially high-resolution soil texture maps are analyzed for their
plausibility, estimator robustness and uncertainty.
The rest of the manuscript is organized as follows: Section 2 introduces the test site, the implemented and auxiliary
data, as well as methods applied and developed. Section 3 shows and discusses the results of the estimator setups
and its cross validations. Finally, section 4 reviews main findings and gives overall conclusions.
**2    Data and Methods**
**2.1    Test site**
The research area for this case study is the Attert catchment (mid-western Luxembourg, Fig. 1), the target site of
the German DFG research project CAOS ("Catchments as Organised Systems") (CAOS, 2015; see also Zehe et
al., 2014). The catchment has a size of 288 km² for the gauge in Bissen and stretches from 222 m to 535 m above
sea level. Collated in former studies (Müller et al., 2014), a spatial dataset containing land cover, geology, elevation
data, and qualitative agricultural soil information is available (see Fig. 2). A schist massif in the north and
sandstone lifts in the far south embrace the undulating central marl area (SGL, 2003). The dominant land use is



agriculture (65.4%), followed by forests (29.7%) and settlements and other sealed areas (4.8%) (Corine land cover;
EEA, 1995). The monthly mean temperature ranges from 0 °C in January to 18 °C in July (1971-2000); the climate
is pluvial oceanic.
The existing agricultural soil map (1:100.000; SPP, 1969; Fig. 2, bottom) lacks of quantitative descriptions but
give hints on spatial patterns of soil texture and its systematic distribution: Silt explicitly occurs in four out of the
six existent soil classes in the area; clay soils are observed in the North West and sandy soils occur in the South
East. Thus, relations between geology and soil can be observed, particularly for schists and clay in the
northwestern, and sandstone and sand in the southeastern region.
**2.2 Soil data**
A number of 212 soil samples were taken within the first 15 cm of the topsoil mineral horizon during different
fieldwork activities throughout the Luxembourg part of the Attert catchment (Fig. 2, lower panel). Project members
from KIT (Karlsruhe Institute of Technology) took 125 out of these samples as undisturbed ring samples of 250
ml volume for various hydro-pedologic analyses and as reference samples for runoff-modeling purposes. The other
87 samples were taken by project members from BOKU/LMU (University of Natural Resources and Life Sciences,
Vienna/Ludwig-Maximilians-Universität, Munich) from 30 sites in an attempt to close spatial and systematic gaps
between the existing sample plots as disturbed ring samples of 250 ml volume for measuring soil texture. The
latter sites were chosen based on agricultural soil classes, geology, and land use information trying to cover the
full spectrum of different classes and the full catchment extent. These sites were sampled multiple times within a
radius of 1 m to achieve information on local uncertainty. The texture samples were analyzed based on sieving
and sedimentation analysis after ISO 11277 in two different laboratories at the KIT and LMU. Due to slight
differences in standards with regard to removal of organic compounds and the use of suspension aids, the data
were linearly homogenized by analyzing eight samples in both laboratories and correcting the small biases with
linear models for the three fractions.
The resulting texture distribution is given in Fig. 3. The samples are visualized within the classification system of
the USDA. The sampled textures consist of merely high silt fractions (mainly 40-60%), lower clay fractions
(mainly 20-40%) and a wide range of sand fractions (0-80%). The dominant soil types are silty clay loam (SiClLo),
loam (Lo) and clay loam (ClLo).
Data from sites with multiple measurements (up to three samples per site) are not aggregated and, hence, include
local uncertainties within a radius of 0.5 m for soil texture. From the multiple sampled sites, an average local
standard deviation of the samples of 4.9 pp for clay fractions can be found; for silt and sand variations are slightly
higher with 7.8 pp, respectively 8.7 pp (overall: 7.1 pp). It is noticeable that the local standard deviation is half the
size of the deviation of the full sample dataset with 8.5 pp for clay, 14.2 pp for silt and 18.9 pp for sand (overall:
16.7 pp).
**2.3 Remote sensing data and deduction of principle components**
We used the ASTER (advanced spaceborne thermal emission and reflection radiometer) instrument on board of
the TERRA satellite (Fujisada, 1995) in course of this study. The satellite has a sun-synchronous orbit, with a





repetition rate from 4 up to 16 days, passing the catchment at 11:40 am CET. As described in Müller et al. (2014),
channel 13 (10.25-10.95 μm) is used exclusively, as thermal signals in this wavelength are least altered by
absorption in the atmosphere. The remote sensing time series then is processed for reaching 15 m resolution geo-
referenced top-of-atmosphere (TOA) temperatures (see Müller et al., 2014, for details).  The used ASTER data set
consists of 28 snow and rather cloud free images from the period January 2001 to June 2012 (Fig. 4, lower panel).
Figure 4 illustrates four exemplary TOA temperature images from the time series with representative patterns for
each season (a-1 to a-4) and the available dates (b). Winter is under-represented with two images, as images with
snow cover but no cloud cover are rare. Half of the images (15) are from spring months; summer and autumn are
represented by six and five images respectively. Based on optical data, it was found that the largest fraction of bare
soil in the area is found in late spring, late summer and early fall (15 images are covering these situations).
Based on the 28 TOA temperature time series data, principle components (PCs; PC1-PC28) are calculated as
described in Müller et al. (2014). The components of a PCA are orthogonal and represent linear independent spatial
patterns. They are ranked by the proportion of explained variance in the original temperature pattern time series,
which gives information on their dominance and stability within the whole time series. The PCs do not contain
specific information about spatial auto-correlation of the patterns but inherit the interior organization from the
thermal time series.
Müller et al. (2014) could show some relation between the most dominant PCs and observable patterns of land use
or geology. The first five most dominant PCs are illustrated exemplarily in Fig. 5; it can be observed that with
increasing numbering, the explained variance and pattern immanent gradients of component values decrease. PC1
and PC2 show similarities to the Corine land cover pattern and geology pattern (Fig. 2). To avoid any influence
from highly modified surfaces where no influence of soil texture on the TOA temperature can be expected,
artificially covered areas (Fig. 2, upper left panel, in red) are cut out from the PCs based on Corine land cover data.
**2.4    Multiple linear regression estimator (MLRE)**
The relationship between soil texture data of the collected soil samples and PCs derived from time series of TIR
data is analyzed by multiple linear regression. Three main steps are executed to provide soil texture maps,
individually for each particle size fraction. First, an automatically parameterized Box-Cox transformation (BCT)
is performed to the particle size fraction data in order to guarantee normality of the residuals (Box and Cox, 1964;
Sakia, 1992; Osborne, 2010; Chun and Griffith, 2013). Then, PCs are chosen based on their significance level (p-
values from F-tests) for the multiple linear regression (MLR). At last, the results of the MLRE are restricted to
values between 0 and 100% for each particle size fraction with a sigmoidal capping function on top of the MLR.
Overall, the MLR is set up by
$$\hat{x}_\lambda = \beta_0 + \beta_1 PC1 + \beta_2 PC2 + \cdots + \beta_n PCn + \epsilon, \tag{1}$$
with the PCs $PC1 \ldots PCn$, the corresponding regression coefficients $\beta_{0\ldots n}$ and residuals $\epsilon$ to estimate the soil texture
fraction $\hat{x}_\lambda$. For an unbiased estimation of $\beta_{0\ldots n}$, the residuals $\epsilon$ have to be normally distributed $N(0, \sigma)$ (Chun and
Griffith, 2013).
The BCT is applied by





$$x'_\lambda = \begin{cases} \frac{x^\lambda - 1}{\lambda}, \lambda \neq 0 \\ \ln(x), \lambda = 0 \end{cases},$$
(2)

where $x$ is the soil data before transformation, $x'$ after transformation and $\lambda$ is a parameter estimated from the data
or error distribution to achieve normality. An optimal $\lambda$ is estimated by an iterative Monte Carlo procedure
allowing $\lambda$ to range between [-5,5] with an accuracy of 0.01 to guarantee finding a global minimum. Tests for
normality of the residuals are executed subsequently.
To restrict the MLR predictions to the natural limits of 0-100% share of a fraction, sigmoidal capping functions
(Franklin, 2013) are used. A sigmoidal function generally is differentiable, with a non-negative ("S") or non-
positive ("Z") first derivative, one minimum and one maximum. The presented approach uses the following
implementation:

$$sig(x) = L + (U - L) * \begin{cases} \left(1 - e^{-\left(\frac{x-sp}{sl}\right)^2}\right), & if \ x \geq sp \\ L, & else \end{cases},$$
(3)

where $L$ and $U$ are the lower and upper limits (here: 0% and 100%), $sp$ is the position of the start of the positive
gradient and $sl$ adapts the steepness of this gradient. These two gradient parameters are optimized for reducing the
root mean square error between corrected regression estimator and sample data as much as possible.
**2.5    Cross-validation**
Cross validation (CV) is a common strategy for the evaluation of model performance and the quantification of
uncertainties (Arlot and Celisse, 2010). CV schemes can differ in the size of training and validation subsamples.
Here, four different CV schemes are applied to analyze potential changes in the uncertainty level resulting from
different sample sizes: First, a simple leave-one-out (LOO) strategy was applied, where all but one sample point
are included for model identification, and the remaining data points are used for model evaluation. This procedure
is repeated so that each point (n=212) is left out once, and model performance in form of the root mean squared
error (RMSE) between measurement and prediction can be calculated.
Three further CV variants are applied to analyze the effect of sample size reduction on the prediction performance.
For this, the sample size is divided into 10%‐ (CV10), 20%‐ (CV20) and 50%‐ (CV50) sized validation subsets
with respectively 90%‐, 80%‐ and 50%‐sized training subsets. The random process of validation-set-generation is
repeated n=212 times in order to have an equal number of evaluations for all CV variants. The performance of the
MLRE prediction during validation is again evaluated using the RMSE for each particle class.






**3 Results**
**3.1 Sample data and soil texture maps**
First, F-tests were performed to evaluate the MLRE model performance with regard to the number of different
PCs considered as regressors, as well as to all possible combinations of PCs (for a given number of regressors).
The p-value of the F-tests represents the probability of the (soil texture) data given the Null-hypothesis ($H_0$) that
all regression coefficients are zero. Low p-values are taken as an overall indication for the "relevance" of the
MLRE. Results showed that significant p-values ($< 0.05$) occurred most often when the first five PCs were
incorporated, whereby a number of three PCs out of the first five components performed best. Lowest p-values for
sand are assessed with a MLR based on PC1, PC2, and PC3; for silt, the most adequate combination is PC2, PC4,
and PC5 and for the determination of clay PC2, PC3, and PC4 are used (given the full data set).
The results of the MLRE are illustrated in Fig. 6. Upper panels show the distribution contours of the 95%- (red)
and 75%-quantiles (blue) for the estimation of different particle fractions, while the lower panels show the
respective distribution of residuals. Sand and clay show higher Pearson correlation coefficients ($r > 0.5$), while silt
is showing a lower value ($r = 0.36$). The optimal $\lambda$-values for the BCT indicate almost normally distributed
residuals for clay, while for silt, the error distribution is skewed left and for sand, the distribution is skewed right.
The overall RMSE for the three texture fraction estimators is 12.7 pp, partitioned into 16.2 pp for sand, 13.0 pp
for silt and 7.1 pp for clay. Hence, the MLR shows a well-defined relation for clay, while sand shows good
correlations with few extreme outliers. For silt the system shows low correlations with a smaller variation of errors.
The MLRE calibrated with the complete field sample set is then used to calculate fully distributed texture maps.
Figure 7 shows the resulting soil texture maps. Each texture fraction is modeled separately with the aforementioned
PC combination. Finally, the three texture fractions are translated into USDA soil types, which are then mapped
back into the catchment (Fig. 7, lower right). A comparison of predicted and observed texture data shows a large
overlap between both (Fig. 7, lower left).
The distribution of soil texture conforms to the distribution of the soil characteristics displayed in the available
qualitative agricultural soil maps (Fig. 2). Clay is dominant in the north, rather sandy soils can be found in the
south and mainly silty soils prevail in the remaining parts of the catchment. Further analysis of the soil texture
distribution reveals relations to topographic structures, different land cover types and geology (Fig. 1 and 2).
**3.2 Cross validation results**
Figure 8 shows the spatial patterns of pixel-based standard deviation of the resulting 212 maps of soil texture
fraction. For each different CV variant as well as for all 212 CV runs, an F-test based selection for the choice of
PCs as described in 3.1 has been performed. The mean texture fraction results from the cross validation runs agree
with the overall result and differences in absolute texture fraction are small when compared to results illustrated
in Fig. 7 (without Figure). The patterns of valley structures and bedrock distribution can be observed here as well.
Statistical indices allow further analysis of the similarities. The spatial average of the coefficient of variation
(c.o.v.;$|\sigma/\mu|$) between all model results, i.e. from CV schemes and the full data set, shows values of below of 0.1





for all texture fractions and underlines the visual impression of small deviations. Supported by the constant RMSE values throughout different variants of the CVs (sand: ~16.6 pp, silt: ~13.4 pp, clay: ~7.2 pp; compare Tab. 1), MLREs indicate a stable behavior in the estimation of spatial texture fraction. The overall RMSE for the different CVs is around 10.9 pp and hence little lower than for the full dataset (12.7 pp). However, silt and sand fractions are higher by ca. 0.5 pp and clay by ca. 0.15 pp.

These values indicate higher uncertainties for silt and sand fractions. This uncertainty is attributed to a higher variation in distribution characteristics within the sample data sets that is increased by decreasing the number of used samples.

In contrast to the stable mean results, textural standard deviation in Fig. 8 varies more. Clay fraction maps show least deviation, while highest deviation is observed in least-sampled sandstone areas in the southeast of the catchment and the lower lands near the gauge. Cross-validation based uncertainties occur as expected: maximum standard deviations rise with increasing size of the validation data sets, and therefore less calibration data.

Table 1: Root mean square error (RMSE) values for the sample data from the different Cross validation (CV) variants by particle size and overall. The results are noted in percentage points [pp]. The different CV variants (LOO, CV10, CV20, and CV50) show similar values for the RMSEs between sample and estimated fractions sand, silt, and clay within the 212 subsets for the CVS (column 2 - 4). Additionally, the approximately constant overall RMSE for all fractions and samples within the CVSs' subsets is noted (column 5).

| CV | RMSE per particle size [pp] | | | overall RMSE [pp] |
|---|---|---|---|---|
| | sand | silt | clay | |
| LOO | 16.58 | 13.37 | 7.18 | 10.80 |
| CV10 | 16.64 | 13.42 | 7.25 | 10.87 |
| CV20 | 16.56 | 13.30 | 7.11 | 10.80 |
| CV50 | 16.89 | 13.54 | 7.34 | 11.00 |

Exceptional high deviations are highly localized throughout the subsets and show areas of higher uncertainty for the estimated texture classes. These outliers are stable, spatially and throughout the CV variants. These deviation hot spots can occur due to soils being out of the range of sampled soil types or specific land cover.

However, the automatically optimized choice of PCs for the regression estimators is quite constant throughout the sample data subsets in the different CV variants. For all three texture classes, PC2 is used by 100% of the estimator setups, the pattern that resembles geology. The main variation within the texture fractions is then added for

    a)   sand by using PC3 (99% of the setups) and PC1 (65%),

    b)   silt by using PC4 (81% of the setups) and PC5 (54%) or PC1 (47%), and

    c)   clay by using PC3 (100% of the setups) and PC4 (84%),





with number of different variants increasing with the size of the validation data. This also hints at a slight
inconsistency within the measured data. Nonetheless, the overall estimator choice seems to be relatively uniform,
especially for clay and sand fractions.
**4   Discussion and conclusion**
This study investigates the potential of estimating distributed soil texture fractions with time series of thermal
remote sensing data. Elementary thermal patterns (PCs) are extracted from the time series with PCA and are used
as inputs in a MLR model framework to estimate soil texture fractions. The MLRE model is calibrated and
evaluated against a set of 212 measured soil texture data using four different cross validation variants. After
calibration, it is applied for the generation of soil texture and soil types maps based on the distributed PCs
information.
The MLRE prediction uncertainties expressed as overall RMSE when using the full data set for calibration is 12.7
pp (sand – silt – clay: 16.2 pp – 13.0 pp – 7.1 pp) and does not change significantly with in different CV variants.
Given local measurement uncertainties of 7.1 pp for all fractions (sand – silt – clay: 8.7 pp – 7.8 pp – 4.9 pp), the
model induced uncertainty component might be in the range of 3 - 8 pp, varying for the different fractions. The
stability of RMSE and choice of PCs reveal the reliability of this simple estimator setup, a distinct relation between
the basic patterns, observed with thermal remote sensing, and the soil texture samples.
A review of different approaches presented in the literature, often using more complex methods or input data,
reveals similar or even higher uncertainties. For example, McBratney et al. (2000) list RMSE for clay content
based on 85 samples within an only 42 ha sized and less heterogeneous area. They test different estimator types in
their work: 7.6-8.2 pp for interpolation methods and 6.2-8.9 pp for regression techniques (regression trees to neural
networks) with combinations of ancillary information (terrain data, yield data, and electromagnetic measurements)
with a resolution of 200 m. The here presented MLRE based on PCs achieves similar uncertainty values for clay
while using less complex methods and input in a more complex terrain. Wang et al. (2015) reach uncertainties of
10.7-15.5 pp for sand and 4.6-6.5 pp for clay, using a regression model on diurnal temperature range data from a
derived land surface remote sensing product with a rather inapplicable coarse resolution of 1 km for a catchment
of 5130 km² with 62 soil samples. The product resulting from the MLRE approach is of higher resolution while
providing similar deviations. Modelling soil texture inversely (40 m) from soil moisture, measured with a passive
microwave radiometer, performed by Santanello et al. (2007) for a watershed of 148 km², leads to RMSE values
of 12-28 pp for sand, 14-25 pp for silt and 0-8 pp for clay fractions. This inversion requires a complex model
(Noah Land surface model; Noah-LSM, 2015), however, results in a much larger range of uncertainty. Overall,
the herein presented estimator method requires the least temporal resolution of remote sensing data and least
amount of additional data, notably none, with a simple regression method. Furthermore, the density of required
measurements is low compared to the mentioned studies.
Our study also demonstrates that extracted PCs from time series of thermal images contain the necessary spatial
information to delineate distributed soil texture information. The different CV variants show that an appropriate
uncertainty range can be obtained with around 100 training samples for roughly 300 km² with a target resolution
of 15 m. Analyzing the deviation maps for the different CV variants (Fig. 8) reveals hotspots of diverging relations



between sample data and PCs. These maps can be used to further reduce uncertainty by localizing effective additional sample locations.

This approach works well within the current resolution of 15 m and with a sparse temporal resolution, but should also work for different setups of spatial and temporal resolution, with a reasonable minimum variation in the time series. Hence, this approach is possible to handle many applications within hill slope, catchment, and up to continental extent with comparably small sampling effort and definite uncertainties.

The presented data can help to improve the generation of topsoil maps, especially without the need of proper soil genesis descriptions. These maps then can be utilized for medium scale catchment setups of eco-hydrological models, especially within (near) ungauged basins. The basic thermal remote sensing time series can also be obtained from other sensors with different resolutions, such as Landsat (60-100 m) or MODIS (1 km). Only few measurements are necessary as long as the spatial extent of the thermal remote sensing data and taken samples cover the statistical distribution of catchment characteristics.

Further applications of this PCA based MLRE to assess spatial distributions of bulk density, topsoil organic matter, vegetation density, or even fraction of absorbed photosynthetic active radiation (FPAR) will be subject to further research.

**Acknowledgements**

We want to thank the German Research Foundation (DFG) and the Austrian Science Fund (FWF) for funding this research through the CAOS (Catchments as Organised Systems) Research Unit (DFG: FOR 1598; Grant SCHU1271/5-1; FWF: I 2142-N29).We also want to thank the LPDAAC (Land Processes Distributed Active Archive Center) for providing free ASTER data. We thank Loes van Schaik, Elisabeth Thiem, Liya Sun, and the student staff for their fieldwork and the laboratory analysis, especially, Thomas Weiss for his work in the LMU laboratory, as well as the Lippmann Research Institute, Luxembourg, for arranging access to the catchment.

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

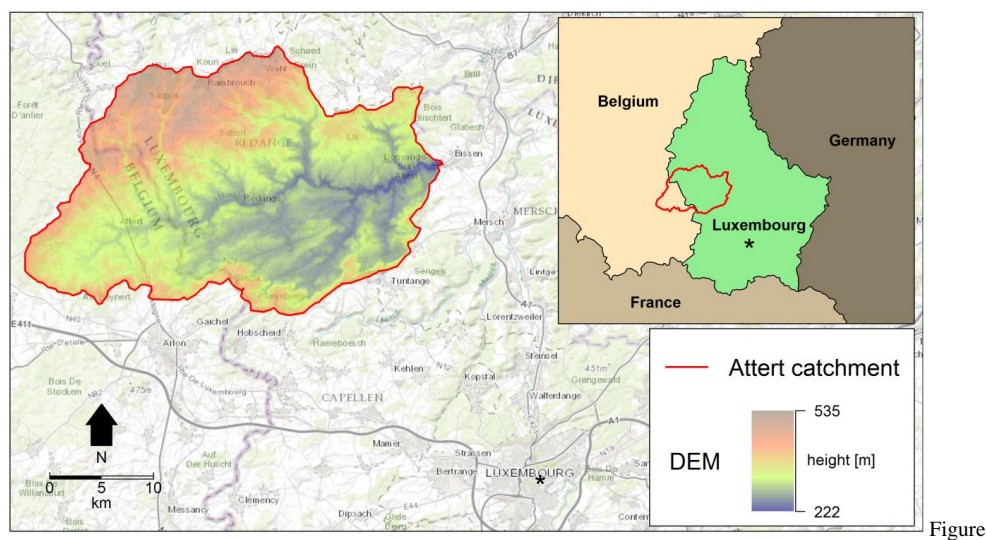

Figure

**Figure 1: The position of the Attert catchment with superimposed elevation data. The gauge Bissen, Luxembourg,**
**defines the catchment boundaries.**

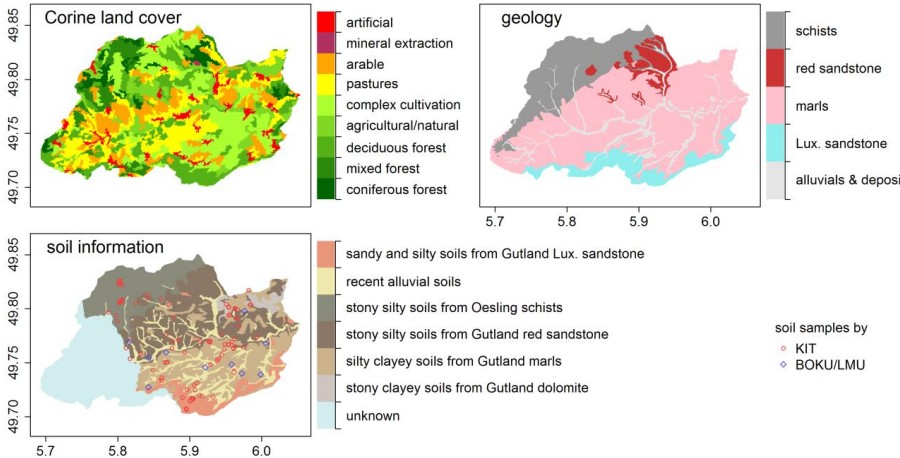

**Figure 2. Corine land cover (upper left), geology data (upper right), and agriculturally relevant soil information (bottom**
**left) of the Attert catchment. The sites of the two sample sets are marked for KIT (Karlsruhe Institute of Technology;**
**red circles) and BOKU/LMU (University of Natural Resources and Life Sciences/Ludwig-Maximilians-Universität;**
**blue diamonds).**





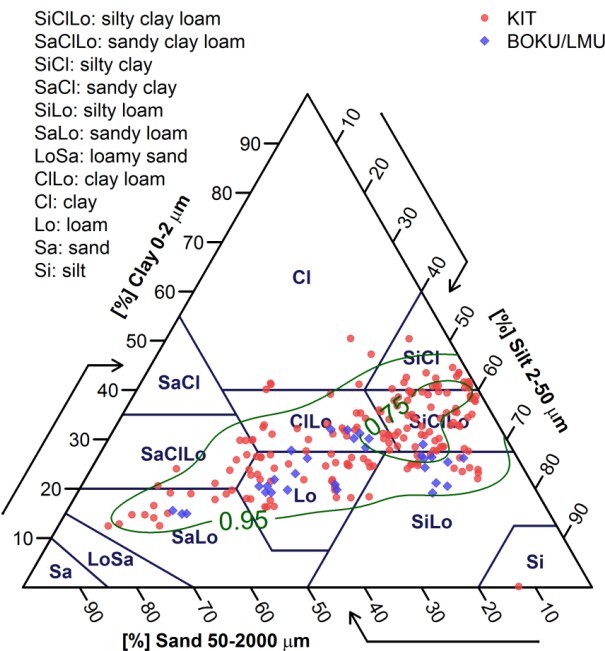

**Figure 3. The categorical distribution of the homogenized soil samples by KIT (red circles), and BOKU/LMU (blue**
**diamonds) is noted within the USDA classification scheme. The symbols are semitransparent to better visualize**
**accumulations. Data from both laboratories overlap properly. Additionally, the estimated probability density is shown**
**in dark green as contours for quantiles of the theoretical distribution of observable soil textures. The 0.95 contour**
**delimits the area, 95% of the textures are lying within; the 0.75 contour defines 75% of the textures, respectively.**



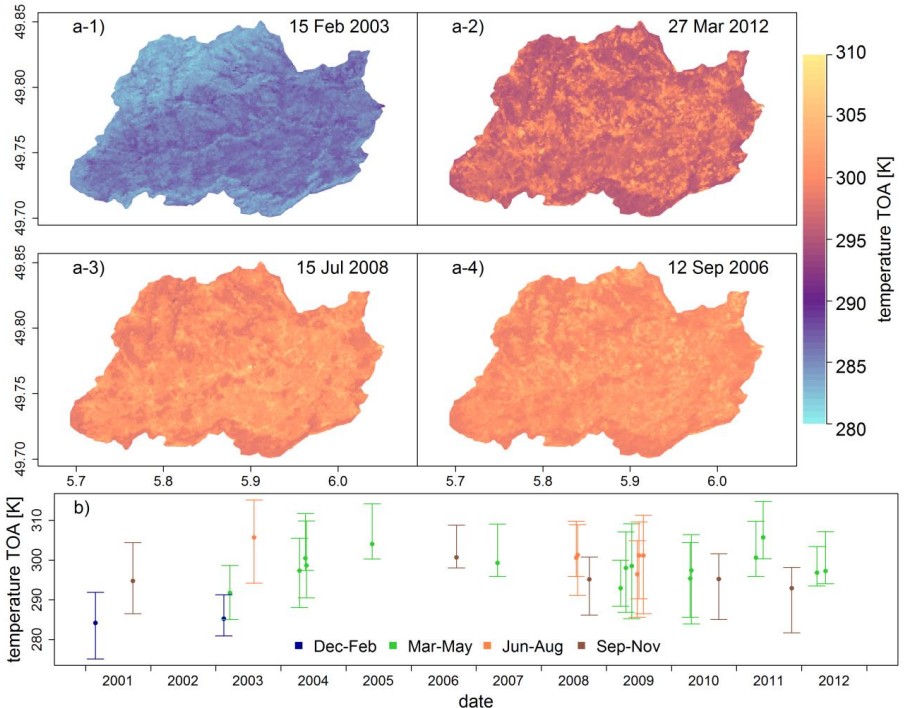

2  **Figure 4.  a) Seasonal examples of the temperature time series patterns (winter 1, spring 2, summer 3, and autumn 4).**

3  **b) Overview of the dates of the TIR image time series some statistical information (mean and absolute ranges).**

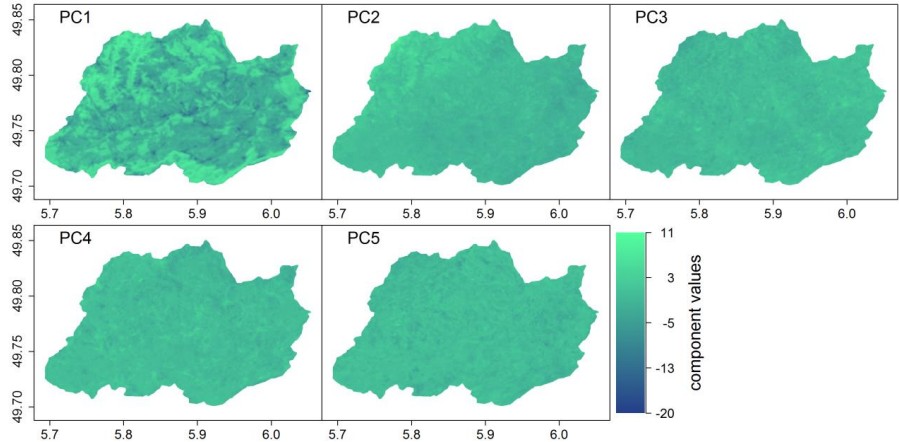

5  **Figure 5. The first five components of the PCA of time series data, sorted by decreasing explained variance.**





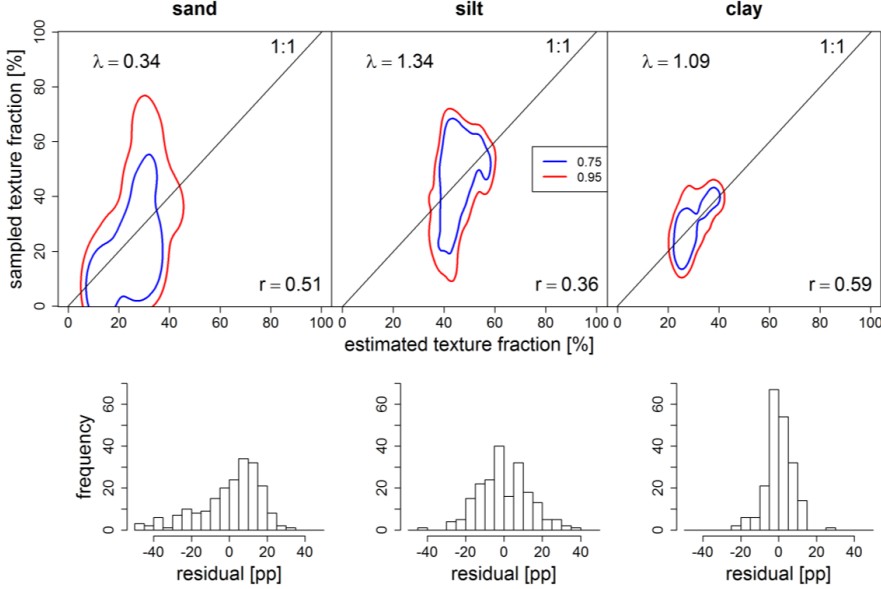

**Figure 6. MLRE results for predicting sand, silt, and clay fractions estimated with three components of the PCA data.**

**The distribution contours (upper panels) show the sampled versus the estimated texture data after BCT and sigmoidal**

**capping. 95%- and 75%-quantiles are shown in red, respectively blue. λ-values of the BCT (upper left) and Pearson**

**correlation coefficient $r$ between the sample and estimator data (lower right) are also noted. The lower panels show the**

**respective distributions of residuals [estimator - data] and their skewness related to the BCT λ-values after back**

**transformation.**





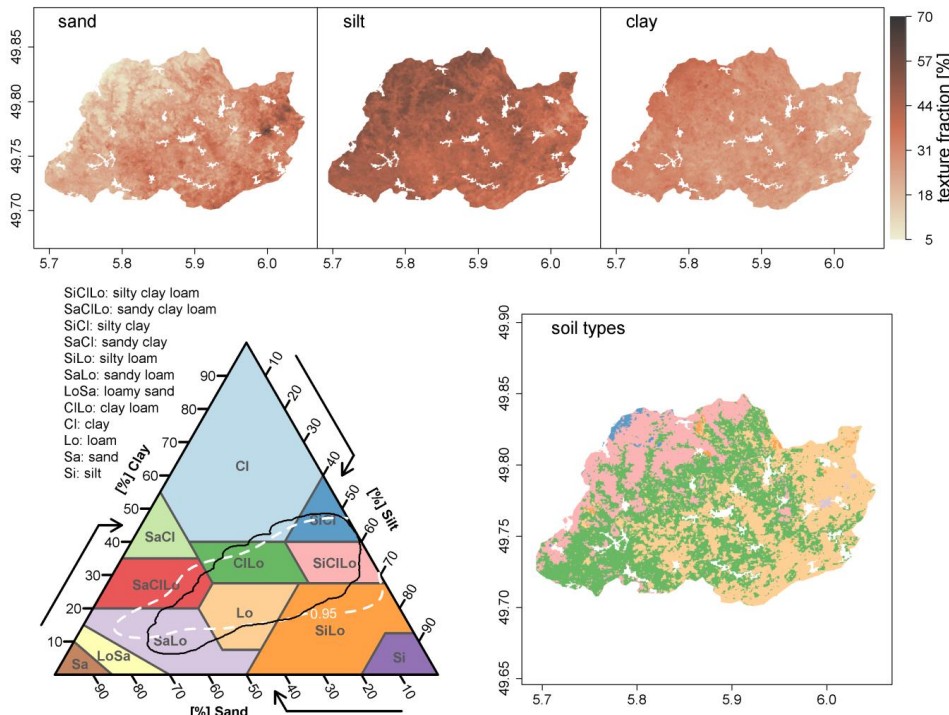

**Figure 7: Result maps for texture fractions for the MLRE predictions based on the full data set are shown (upper**
**panels). Lower left: Texture triangle with the resulting data distribution (black outline) compared to the 95%-quantile**
**distribution estimation of the measured samples (dashed white) from Fig. 4, providing the color legend for the soil type**
**map (lower right). Therein, artificial areas (Corine land cover) are ignored (white).**





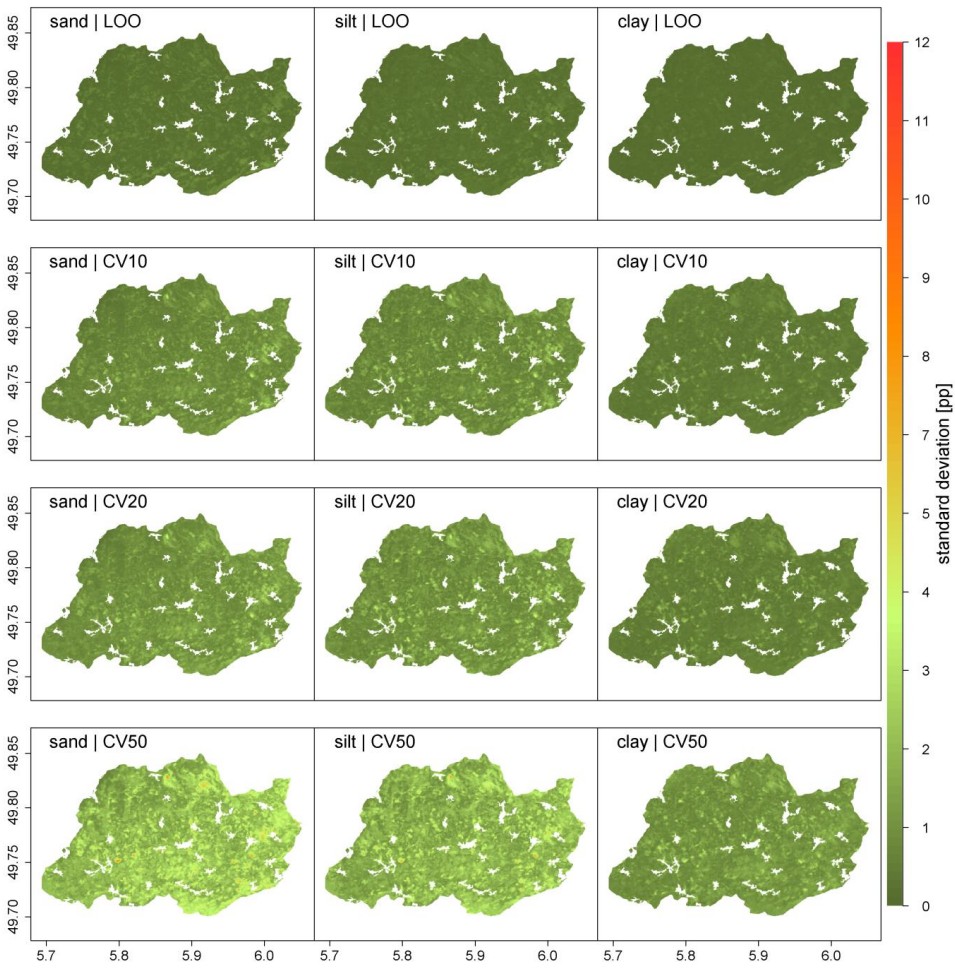

2   **Figure 8. Spatial patterns of standard deviations for the different CV variants (rows) of soil texture fractions (columns)**

3   **calculated from the estimators for the 212 different randomly selected data sets. Hotspots of high uncertainty (5 pp and**

4   **above) can only be observed with CV50 subsets and within sand and silt fractions.**

