# Peer review of "Estimating distributed soil texture using time series of thermal 1"

_Hydrology and Earth System Sciences, 2016_

## Referee Comment (RC1) · Anonymous Referee #1 · 20 Apr 2016

The authors tried to estimate the soil texture information from the thermal remote sensing observation, using PCA method and MLRE estimator, and soil sample data. Such approach is innovative and worthy of further exploration of its application for other type of data, for example, microwave or optical RS data. It seems to me this approach is more generic and can be applied for different types of remote sensing observations. Nevertheless, i do have some comments for the authors to consider, for a better presentation of their works.

Major concerns: 1. As said, it seems this approach can be applied for other type of RS observations (i.e. microwave or optical). Although the author mentioned the work done by Santanello et al. 2007 in the discussion, their approach is completely

different (e.g. using Noah model) from the current study. The author is encouraged to do analysis on how the current proposed PCA+MLRE approach can be applied using microwave/optical RS data.

2. In the study area, the silt predominates, yet the uncertainty associated with the estimation of silt is the largest. There is no detailed discussion on why such uncertainty. Are there any further investigation on how many soil samples over silt area? or some analysis from thermal inertia over silt? In addition, it would be better to indicate explicitly the percentage of areas of the test site, in terms of different soil texture, from the estimate and the available in-situ data (e.g. % sand area, % silt area, etc.).

3. In the introduction, the authors argued that the choice of thermal RS is due to the thermal inertia. The thermal inertia can be derived from RS images with Anne Verhoef's approach. I am wondering if the PCA+MLRE was applied with the thermal inertia time series, would the results be different or similar? Please also provide some discussions on this point.

4. Some presentation of the results are not detailed and, therefore, easy to cause confusions for readers. I will detail this point with minor comments as following.

Minor comments: 1. Page 4, line 19-20, "PC1 and PC2 show similarities to the Corine land cover pattern and geology pattern"

It is not obvious from the Figures. And, i cannot figure out how and why they are similar, and what does it mean by this similarity? Please quantify such similarity using more detailed information.

2. Page 6, line 24-25, here you compared the estimated distribution of soil texture (Fig.7) with the available qualitative agricultural soil map (Fig. 2). However, such comparison for readers is difficult, as the color legend are different from the two figures. Is it possible to unify the color legend of the two to enable the direct comparison?

3. Page 6, line 26-27, "Further analysis of the soil texture distribution reveals relations

to topographic structures, different land cover types and geology (Figure 1 and 2)". Just by saying that, i cannot find any further analysis on this point in the text. Please provide them with detailed analysis. This will assist the readers to understand straightforwardly.

4. Page 6, line 31-33 "...... in Fig. 7 (without figure)", i was lost here.

5. Page 7, line 22-23. "These outliers are stable, spatially and throughout the CV variants ....". I cannot find them. It is not obvious. Please use circles to highlight those hot spots.

6. Page 8, Line 5-6. Here you used the time series of thermal remote sensing data. What about using microwave/optical data as well? Please add some more discussion on this aspect here.

7. Page 15, Figure 7 caption, line 4, ' ... are shown in red, respectively blue"?

---

## Referee Comment (RC2) · Anonymous Referee #2 · 7 May 2016

Review result for Journal:     Hydrol. Earth Syst. Sci. Discuss.

Manuscript Number          HESS-2016-115

Title:                                    Estimating distributed soil texture using time series of thermal
                                           remote sensing – A case study in central Europe

The paper adresses an important topic and indicates valueable research results due to the estimation of soil texture data using thermal satellite remote sensing data. The manuscript presents relevant research information and is easy to read and to understand. I believe that with minor revisions the manuscript has potential to be a real scientific contribution and cover a timely issue for a broad readership. I suggest the manuscript for publication in HESS.

Reviewers Suggestions:
   - I suggest to insert the expression „spatial or spatially"  (e.g. line 7 and title) to soil texture pattern
   - Insert a conclusion sentence at the end of the abstract that gives information about the use of the research results or outlook for future developments in remote sensing and hydrology community.
   - I recommend short explaining sentences on the used remote sensing TOA signal data as it is applied as physical source of information about soil texture.

   - English and grammar: I provide not the proper competence to review language errors.

---

## Author Comment (AC1) · 21 Jun 2016

RC1.1) Our approach can be generally applied to any remote sensing time series that can be related to soil hydrological parameters. Yet,no data base for further analysis is compiled, but future research will try to evaluate differences to and similarities with other sensors. A mixed data base will also be evaluated.

RC1.2) The original pedological map is based on qualitative data and extremely vague in its description. The number of dominantly silty samples can be estimated from Figure 3. Actually, a certain amount of silt is existent in all of the soil samples. The observed high uncertainty is a consequence of small thermal differences between soils of widely different silt fractions. This issue was objected in Wang et al. (2015). We will add a

short paragraph which will address this issue.

RC1.3) Thank you for pointing to Verhoef's work (Murray and Verhoef, 2007). They are using excessive knowledge on soil hydrological parameters. Our approach is actually using none. For a thermal inertia time series, we would need spatially distributed information about soil hydraulic parameterizations which is currently non-existent. Thus, we cannot directly compare a calculated thermal inertia time series. We will add a paragraph to the discussion section, concerning this aspect.

RC1.4) 1) We will add reference to Müller et al. (2014) for further details on the comparability. 2) Sadly, it is not possible to unify the mentioned color legends. These are different as the contents of the maps differ. We intend to stay with the current coloring to exclude misconceptions. 3) We will extend the existing explanation. A complete quantitative explanation would go beyond the scope of this paper. 4) We will include a figure (Fig. R1). 5) We will exchange Fig. 8 with Fig. R2. 6) We will extend the specific paragraph. 7) We will change the text to "...95%- (red outline) and 75%-quantiles (blue outline) are shown."

---

## Author Comment (AC2) · 21 Jun 2016

RC2.1) We will add "spatial" to these expressions.

RC2.2) We will add a corresponding last sentence. Thank you for this suggestion.

RC2.3) We tried to reduce text by referring to Müller et al. (2014). The preprocessing and description of the TOA data is explained there to a further extent. Our feeling was that a recapitulation is beyond the scope of this paper.

Additional references:

Murray T., Verhoef A., 2007: Moving towards a more mechanistic approach in the

determination of soil heat flux from remote measurements: I. A universal approach to calculate thermal inertia, Agricultural and Forest Meteorology, Volume 147, Issues 1–2. doi:10.1016/j.agrformet.2007.07.004.

Wang DC, Zhang GL, Zhao MS, Pan XZ, Zhao YG, et al., 2015: Retrieval and Mapping of Soil Texture Based on Land Surface Diurnal Temperature Range Data from MODIS, PLoS ONE 10(6): e0129977. doi: 10.1371/journal.pone.0129977

––––––––––––––––––––––––

---

## Author Comment (AC3) · 21 Jun 2016

The following figures will be included in accordance to Reviewer1.

Figure R1: Exemplary mean value maps of soil texture fractions estimated with the CV50 CVS from the possible 212 different data sets. There are no significant differences between the different mean value maps for the different CVSs. Moreover, differences to the maps based on the full sample set are minor (Fig. 7).

Figure R2: Spatial patterns of standard deviations for the different CVSs (rows) of soil texture fractions (columns) calculated from the estimators for the 212 different randomly selected data sets. Hotspots of high uncertainty (5 pp and above; exemplary spots are

[Figure]

circled on all 12 maps) can only be observed with CV50 subsets and within sand and silt fractions.

—————————————————

[Figure]

[Figure]

**Fig. 1.**

[Figure]

**Fig. 2.**

---

## Author Response (AR1)

**Reply to Reviewer 1:**

RC1.1) Our approach can be generally applied to any remote sensing time series that can be related to soil hydrological parameters. Yet, no data base for further analysis is compiled, but future research will try to evaluate differences to and similarities with other sensors. A mixed data base will also be evaluated. A short overview over potential application is given at page 15.

RC1.2) The original pedological map is based on qualitative data and extremely vague in its description. The number of dominantly silty samples can be estimated from Figure 3. Actually, a certain amount of silt is existent in all of the soil samples. The observed high uncertainty is a consequence of small thermal differences between soils of widely different silt fractions. This issue was objected in Wang et al. (2015). We have clarified the connection between our work and Wang by citing their work at the respective paragraphs (e.g. Page 10).

RC1.3) Thank you for pointing to Verhoef's work (Murray and Verhoef, 2007). They are using excessive knowledge on soil hydrological parameters. Our approach is actually using none. For a thermal inertia time series, we would need spatially distributed in- formation about soil hydraulic parameterizations which is currently non-existent. Thus, we cannot directly compare a calculated thermal inertia time series. We have added a paragraph which is discussing the connection to their work (Page 15)

RC1.4) 1) We will add reference to Müller et al. (2014) for further details on the com- parability. 2) Sadly, it is not possible to unify the mentioned color legends. These are different as the contents of the maps differ. We intend to stay with the current coloring to exclude misconceptions. 3) We have extended the existing explanation. A complete quantitative explanation would go beyond the scope of this paper. 4) We have included another figure (8) in accordance to the Reviewers suggestions. 5) We have overworked Fig. 9 (former figure 8) in accordance to the Reviewer. 6) We have extended the specific paragraph (Page 13) in accordance to the Reviewer. 7) We have changed the text to "...95%- (red outline) and 75%-quantiles (blue outline) are shown."

**Reply to Reviewer 2:**

RC2.1) We have added "spatial" to these expressions (e.g. title; page 2 and so on).

RC2.2) We have added a corresponding last sentence. Thank you for this suggestion.

RC2.3) We tried to reduce text by referring to Müller et al. (2014). The preprocessing and description of the TOA data is explained there to a further extent. Our feeling was that a recapitulation is beyond the scope of this paper.

Additional added references:

[revised manuscript text omitted]